# Near-Infrared On-Site Evaluation (NOSE) Examination of EBUS/EUSb Samples—A New Method for Sample Adequacy Evaluation

**DOI:** 10.3390/diagnostics14171887

**Published:** 2024-08-28

**Authors:** Jiri Votruba, Ivan Čavarga, Tomas Bruha, Zuzana Sestakova

**Affiliations:** 11st Clinic of Tuberculosis and Respiratory Diseases, General University Hospital in Prague, U Nemocnice 2, Prague 2, 120 00 Prague, Czech Republic; zuzana.sestakova@vfn.cz; 2Pulmonary and Phtisiology Clinic, Bratislava University Hospital Ruzinov, Ruzinovska 6, 82606 Bratislava, Slovakia; cavarga.ivan@gmail.com

**Keywords:** fine-needle aspiration biopsy, NIR radiation, EBUS/EUSb specimen, spectral absorbance

## Abstract

Fine-needle aspiration biopsy is crucial for modern diagnostics of endoscopic procedures and thus an efficient and reliable method for increasing biopsy yields is urgently needed. In our study, we address the limited availability and high price of the rapid onsite evaluation (ROSE) technique by introducing the technique of near-infrared on-site evaluation (NOSE) consisting of spectral measurement of near-infrared radiation (NIR) transmitted through the evaluated material. For this purpose, we designed a special optical probe consisting of two fibres, of which one is a source fibre and the second is a detector fibre. The distal ends of both fibres are brought together into one bundle which is, with the help of a special extension, applied to a cuvette with an analysed sample at a defined distance from the cuvette bottom and fixed in place. A portion of the NIR radiation received by the detector fibre after it propagates through the sample then depends on the optical and therefore morphological characteristics of the sample. Based on the measured spectral curve, we can calculate the attenuation coefficient curve and subsequently the parameter of the sample richness and the parameter characterising the autofluorescence peak as well. We found that the value of our introduced parameters is in significant relation to sample richness as well as to sample malignity. NOSE evaluation of EBUS/EUSb (endobronchial/oesophageal ultrasound bronchoscopy) specimens can be considered an easy new technique aiming to improve sampling diagnostic accuracy and to diminish costs related to the presence of a cytopathologist and related instrumentation in the endoscopy suite.

## 1. Introduction

Fine-needle aspiration biopsy is crucial for modern diagnostics in a wide range of endoscopic and trans-cutaneous biopsy procedures. Linear endobronchial ultrasound (EBUS) has revolutionized the field of respiratory medicine by providing a minimally invasive and highly accurate method for diagnosing and staging lung diseases. This technology combines bronchoscopy with real-time ultrasound imaging, allowing for real-time visualization of the airways and surrounding structures.

One of the primary applications of linear EBUS is the diagnosis of lung cancer. By using the ultrasound probe to guide the bronchoscope to suspicious lesions, it is possible to obtain tissue samples for histological analysis without the need for more invasive procedures. This not only streamlines the diagnostic process but also reduces the risks associated with surgical biopsies. In addition to diagnosis, linear EBUS plays a crucial role in the staging of lung cancer. By providing detailed imaging of the mediastinal lymph nodes, EBUS accurately determines the extent of the disease and selects the most appropriate treatment option for each patient. This information is essential for optimizing patient outcomes and guiding therapeutic decision-making. Furthermore, linear EBUS is a versatile tool that can also be used to diagnose other lung conditions, such as sarcoidosis, tuberculosis, and infections. The ability to visualize and biopsy suspicious lesions in the airways and lymph nodes provides valuable diagnostic information that can guide treatment planning. Compared to surgical procedures, EBUS is associated with lower complication rates, shorter recovery times, and reduced healthcare costs. Patients benefit from the convenience of undergoing the procedure on an outpatient basis, minimizing disruptions to their daily lives. As technology continues to advance, the capabilities of linear EBUS are expected to expand even further.

Ongoing research and development are enhancing the resolution and image quality of EBUS systems, making it easier to identify and biopsy small or subtle pathologies in the hilar and mediastinal/paraesophageal areas.

Examination of the biopsy sample has also developed rapidly recently.

The methodology itself uses several optimization procedures to increase biopsy yields, such as different biopsy needle designs, aspiration using negative pressure, number of punctures, and the macroscopic on-site evaluation (MOSE) or the rapid on-site evaluation (ROSE) technique [1,2,3]. During the endoscopy procedure, it is important to determine when to stop the biopsy process because we assume that the samples are adequate for obtaining histology [4]. This has an impact on both patient safety (length of procedure, repeated punctures) and economy (needle consumption, time of endoscopy team at work). ROSE and MOSE techniques have been widely used to help with finding the ideal time to stop the biopsy procedure. The availability of the rapid on-site evaluation technique is often problematic, as a cytopathologist is not immediately available in endoscopic rooms. A promising solution to this problem is the possibility of a trained endoscopist examining the cytology smear themselves as part of the first opinion, which is later confirmed by the pathologist [5]. Overcoming limited availability of the ROSE technique was addressed by Iwashita and coworkers [6] by introducing a procedure for macroscopic evaluation of specimens obtained during EUS biopsy of solid pancreatic lesions. The authors deposited specimens from fine-needle biopsies on a glass slide by extruding the stylet from the lumen of the needle. They carefully evaluated the specimens, looking for macroscopically visible tissue cores. Tissue cores were defined as white- or yellow-coloured pieces of apparently more rigid consistency as opposed to the paste-like or liquid-like areas of the biopsy specimens. Out of a total of 237 specimens, macroscopic tissue core was found in 216 specimens (91.1%). Blood clotting was confirmed in 171 specimens (72.2%). Histological yield of biopsy was observed in 187/237 specimens with white or yellow tissue core findings / 78.9%/, but blood clots were found only in 17/237 specimens (9.3%). The authors determined the optimal length of the visible tissue core to be more than 4 mm. In this group, the sensitivity and specificity for obtaining a histological diagnosis from the tissue core were 93.1% and 72.0%, respectively [6]. This pilot study was followed by several confirmatory studies. Ishikawa et al. [4] compared the MOSE and ROSE techniques using almost equivalent optical technologies, namely, a microscope for evaluation of cytologic smears in ROSE and a stereomicroscope with magnification of 7× to 90× in MOSE. The gain in histologic diagnosis was evaluated. According to the criterion evaluation for MOSE, 90% of the specimens (54/60) had white core tissue and were judged positive, and 10% (6/60) had weak or no core tissue and were judged negative. Evaluation by the MOSE technique was able to correctly predict the positivity of the histological diagnosis in 54 of 60 cases. In this study, MOSE showed higher sensitivity (98% vs. 68%) but lower specificity (60% vs. 50%) than ROSE. Both methods had similarly high clinical utility once the main purpose of the two procedures is established. Another work evaluated the usefulness of a newly developed system which automatically evaluates the total amount of whitish cores in endoscopic ultrasound-guided fine-needle aspiration biopsy (EUS-FNAB) samples using an automated multi-band imaging system. The method utilized different spectral absorption characteristics of blood coagulate when a multi-band LED light source having peak wavelengths of 405, 430, 465, 505, 545, 600, 630, 660, and 700 nm was applied [7]. A randomized controlled trial confirmed that in the absence of ROSE, EUS needle aspiration with the MOSE technique could achieve a diagnostic yield comparable to that of conventional technique with fewer passes [8]. In the study performed with endobronchial ultrasound fine-needle aspiration the presence of white material in the biopsy specimen also correlated strongly with specimen adequacy [9].

Recently, various machine learning methods have also been applied with the goal of categorizing objects (samples) into distinguished classes based on similar properties or attributes. The system mainly relies on pattern recognition (including computer-aided diagnosis (CAD). CAD system performance is very promising in the area of pattern recognition during optical measurement of biopsied tissues.

In our work, we address the limited availability and high price of the ROSE technique using NOSE (near-infrared on-site evaluation). The technology is based on instant ex vivo measurement of EBUS/EUS material by near-infrared spectroscopy.

Near-infrared radiation (NIR) is defined as light with wavelengths from 700 to 2500 nm. Such radiation passes fairly easily through several centimetres of biological samples, due to its relatively low absorbance. Its interaction with biological tissue can be described as follows: the radiant flux coming to the tissue surface is subsequently scattered in the tissue, while another part is absorbed in the tissue, part is reflected back, and a portion is transmitted through it. The ratio between the transmitted radiant flux Φ_B_ and total radial flux coming to the surface Φ_S_ is transmittance coefficient τ = Φ_B_/Φ_S_ or as the spectral transmittance coefficient τ_λ_ = Φ_Bλ_/Φ_Sλ_ respectively, for a certain wavelength.

As can be seen, this transmittance coefficient depends on the optical characteristics of the tissue being investigated, represented by the absorption coefficient *μ^a^_λ_* scattering coefficient *μ^s^_λ_*, and the index of refraction *g*, as well as on anisotropy, with respect to the isotropy properties of the tissue. It can be assumed that the aforementioned optical characteristics are in strong correlation with the morphological characteristics of the tissue.

## 2. Materials and Methods

We have designed special measuring equipment for the purpose of spectral measurement of NIR transmitted through evaluated material (cytosediment) in a cuvette. This equipment consists of an optical probe, a special extension for the fixation of the probe at the distal end of the cuvette, a light source, and a spectroscope with a notebook. The optical probe consists of two optical plastic fibres 1 mm in diameter and equipped at the proximal end with a standard SMA 905 connector to enable them to connect to a light source or spectroscope. These fibres converge at the distal ends in one bundle and are covered with insulating sleeving so that the outer diameter of the probe is certain to be applied inside the cuvette with the help of the special extension. One of the fibres is the source fibre and is connected at the proximal end to the light source, while the second is the detector fibre and is connected to the spectroscope. The measuring distal end of the detector fibre is 0.3 cm longer than the source fibre, and it is separately covered with insulation all the way to its end. The ending is cut at an angle of 60 degrees and titan-coated in order to facilitate NIR light absorption from the source fibre. In order to ensure the position of the distal end of the probe remains fixed within the cuvette, we have used a specially designed extension applied to the cuvette. As light source, we have used an Avantes Avalight halogen light, which transmits within a spectral interval of 550–2500 nm. Concerning the spectrometer, we have used a model Avaspec-ULS-TEC air-cooled spectrometer manufactured by the Avantes comp, Apeldoorn, Netherlands as well. The working principle of the described equipment is as follows: Electromagnetic radiation (light) is transported from the light source via the source fibre to the analysed sample within the cuvette. The described experimental set-up is illustrated in Figure 1.While the light penetrates the analysed material, scattering, reflection, autofluorescence, and absorption may occur (depending on the concentration of specific substances within the sample). This results in the specific spectral signal of the radiation being absorbed by the distal end of the detector fibre, which can be visualized with the help of a spectroscope connected to a notebook. For spectral evaluation, we used a major part of the so-called optical diagnostic window [10], defined as the wavelength interval of electromagnetic radiation where haemoglobin’s attenuation is significantly decreased to between 638 and 900 nm (Figure 2). As seen, this set-up allowed us to detect the real-time (on-site) spectral curve of NIR after its penetration through an analysed sample. Based on such a spectral curve, we can obtain information about sample density (signal attenuation) and cellularity/tumorous content (manifested, for example, as an autofluorescence peak).

A description of the measuring procedure follows: Initially after the insertion of the distal end of probe into the extension, we record the white reference signal as a signal reflected from a white surface, as well as a dark reference signal (for potential noise detection). Next, after every EBUS/EUSb procedure, when the cuvette is filled with nodal puncture content obtained from medical personnel along with information regarding the puncture number, the optical probe with its applied special extension is inserted into the cuvette and properly positioned. (i.e., the distal end of the probe is aligned with the cuvette’s vertical axis at a defined distance from the cuvette bottom). Subsequently, the spectral curve is recorded with its appropriate puncture number—this process takes approximately 10 s—then the optical probe is taken out, the distal end is cleared and disinfected, and the cuvette is returned to medical personnel. Subsequently, a mathematical operation is applied to the obtained spectral data in order to obtain the corresponding R, D_AFC,_ and SPR values. This procedure takes approximately 1 min and, according to results, the relevant puncture is declared sufficient or the next puncture is performed. The next step follows after the histology evaluation of the sample when the obtained results are subsequently correlated with the result of our measurement, mainly AF peak characteristics. There is no need for special training of bronchoscopy staff, just short instructions on how to clean and disinfect the distal end of the probe after measurement without risk of probe damage.

Initially we recorded the reference spectrum only from a blood/water mix (i.e., tissue-free sample), representing “zero” punctures. Subsequently, for the puncture sample spectral curve, we used the following algorithm to calculate the significant parameter of sample richness (SPR):(a)We calculated the relative absorption (reciprocal value of transmittance T) from the white signal as the ratio of the intensity of the white reference signal and the puncture sample signal.(b)Subsequently, we used the Beer–Lambert law, which is generally used for the determination of chemical species concentration within light-absorbing materials with the help of the calculating attenuation coefficient as follows:
Ɛ = ln(1/T)/d (cm^−1^)(1)
where d is the optical path length, which is in our case equal to the distance between the distal end of the detector and the source fibre (d = 0.3 cm).

(c)Within the analysed interval 638–900 nm, we calculated the ratio

R = Ɛ _max_ / Ɛ ref_max_
(2)
(where Ɛ ref_max_ is the max. attenuation coefficient referencing the blood/water mix spectrum resulting in a value of 7.513).

(d)We detected the puncture sample spectral curve significant peak (or in the case of attenuation spectra, the significant valley) within a wavelength interval of 698–719 nm, so we concluded that this one can be attributed to an autofluorescence (AF) signal, which means that its height/depth can be also be important indicators of tissue content within the sample. In order to obtain the SPR value, we added up to the previously calculated R, sum of the depth of the AF valley from both sides (see detail in Figure 3).

D _AF c =_ D_AF 1 +_ D_AF 2_
(3)

**Figure 3 diagnostics-14-01887-f003:**
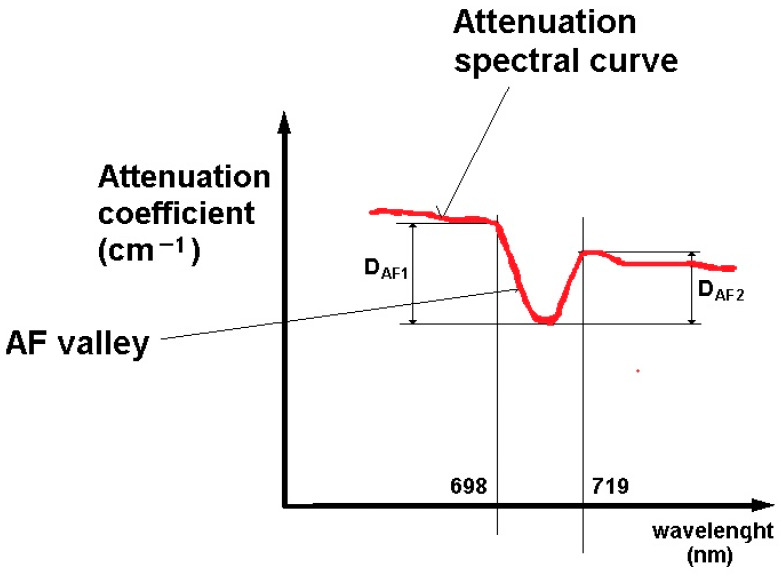
Detail of AF valley within attenuation spectral curve.

(e)Finally,

SPR = R + D_AF C_(4)
where the SPR reference value for “zero” puncture results in 1.015.

In order to detect the correlation of the NIR spectral response with pathological types (especially malignant tissue AF responses) of punctured tissue, we divided the sampling material for purpose of evaluation to four major groups as follows:Tumours + lymphoma (malignant material)Lymphocytes, nodule cells, and anthracotic nodule (non-malignant material)Few nodule cells (poor material)Sarcoidosis (interstitial processes)

Subsequently, we calculated the mean values of R, D_AF c_ and SPR parameters with standard deviation (SD) for each group, and finally we performed a T-test of significance difference for means of SPR and D _AF c_ group values, always between two groups, as follows: 1 to 3, 1 to 2, 2 to 3 and 4 to 2.

EBUS/EUSb examination was conducted as described elsewhere [11]. Study has been designed as prospective non-randomised evaluation of 60 consecutive EUB/EUSb examinations from a non-selected population of patients at a tertial centre endoscopy unit. The indication criteria and contraindication criteria coincided with those associated with the diagnostic process of mediastinal/hilar adenopathy [11]

## 3. Results

A total of 60 consecutive EBUS procedures were conducted at one bronchoscopy centre from 11/22 to 2/23. After every EBUS/EUSb pass, the content of the cuvette was examined by NIR spectroscopy as described above, and obtained spectra were processed. Figure 4 shows for each group the parameter, median, max + min value with 25% and 75% quartile.

Appendix A contains the whole set of acquired SPR, R, and D_AFC_ values; Table 1 contains the calculated means with the standard deviation of final puncture SPR, R and D_AFC_ values for individual groups of detected pathologies types; and Table 2 illustrates the results of statistical comparison between selected groups according to Table 1.

As we obtained from each nodal investigation the relation of its SPR value to its serial puncture number, we found that in most cases, this relation meets the criteria for a linear or logarithmic trend line with a satisfying value of correlation coefficient of fit dependency (graph samples on Figure 5a,b). It should be noted that in cases where logarithmic trend sufficient (maximal) richness was achieved and in cases of linear trends, the puncture sample can be still richer. It can be stated that a resulting arbitrary value of PR of more than 2.5 is predictive of a high-quality sample and achieving that value resulted in a 98% diagnostic yield in a histology/cytosediment evaluation.

## 4. Discussion

Rapid on-site evaluation (ROSE) of biopsy material nodal puncture is a valuable tool that has the potential to revolutionize the diagnostic process for patients undergoing endoscopic nodal biopsy. This technique allows practitioners to quickly assess the adequacy and quality of the sampled tissue, leading to more accurate and timely diagnoses, and ultimately improving patient outcomes. By quickly assessing the sampled material on-site, we can determine whether a sufficient sample has been obtained for accurate diagnosis. This immediate feedback allows adjustments to be made during the procedure, ensuring that a high-quality sample is collected and reducing the likelihood of inconclusive results or the need for repeat biopsies. NOSE evaluation of EBUS/EUSb specimens is an easy new technique aiming to improve sampling diagnostic accuracy and to diminish costs related to the presence of a cytopathologist and its instrumentation in the endoscopy suite. Our results show good clinical applicability as our introduced parameter of sample richness show clear linear or logarithmic ascend tendency with increasing numbers of punctures (as seen in Figure 5)

Furthermore, as seen from Table 2 and Figure 4, malignant material (group 1) expresses significantly higher values of SPR and D_AFC_ compared to sub-optimal sampling—poor (group 3) and non-malignant material (group 2) whose difference of SPR and D_AFC_ is mutually non-significant) and similarly sarcoidosis (group 4) express only significantly higher SPR values compared to non-malignant material (group 2). We can also conclude that our measurements proved significant autofluorescence of malignant material compared to all other types of material. NOSE examination of EBUS/EUSb material seems to be a simple and inexpensive means to overcome the main weakness of ROSE cytology evaluation, which is that it is necessary to have a pathologist on-site, as well as unnecessary delays in EBUS/EUS/b diagnostics. Such evaluation can help prevent the collection of suboptimal samples that may yield inaccurate or unreliable results, ultimately leading to more precise diagnoses and treatment decisions. New possibilities of rapid on-site evaluation of biopsy material from nodal puncture are promising. In comparison with currently used techniques our approach is quicker, does not require so much preparation and effort, and is much more easily repeatable. This technique has the potential to improve the accuracy and quality of diagnoses, expedite the diagnostic process, and enhance patient care.

By incorporating NOSE into endoscopic nodal biopsy procedures, we can help patients to receive the most timely and accurate diagnoses possible.

## Figures and Tables

**Figure 1 diagnostics-14-01887-f001:**
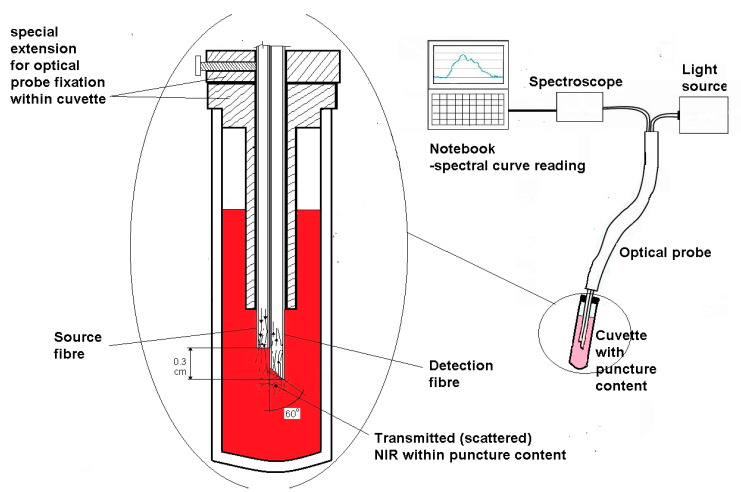
Experimental setup.

**Figure 2 diagnostics-14-01887-f002:**
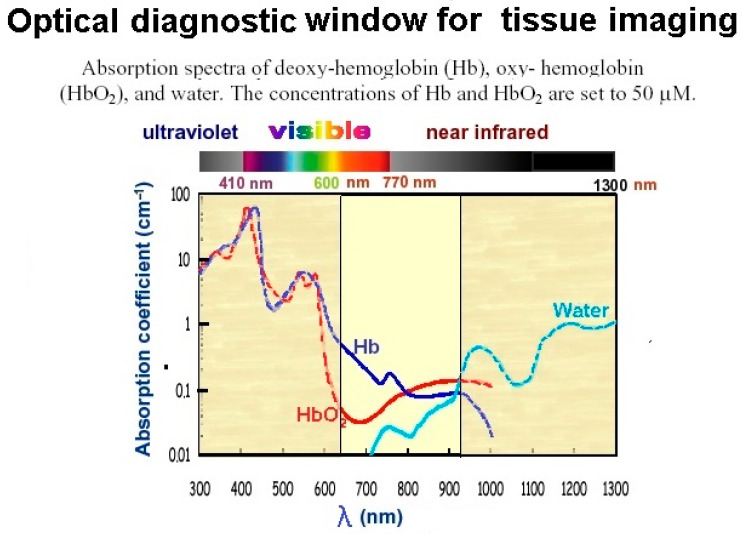
Optical diagnostic window [10].

**Figure 4 diagnostics-14-01887-f004:**
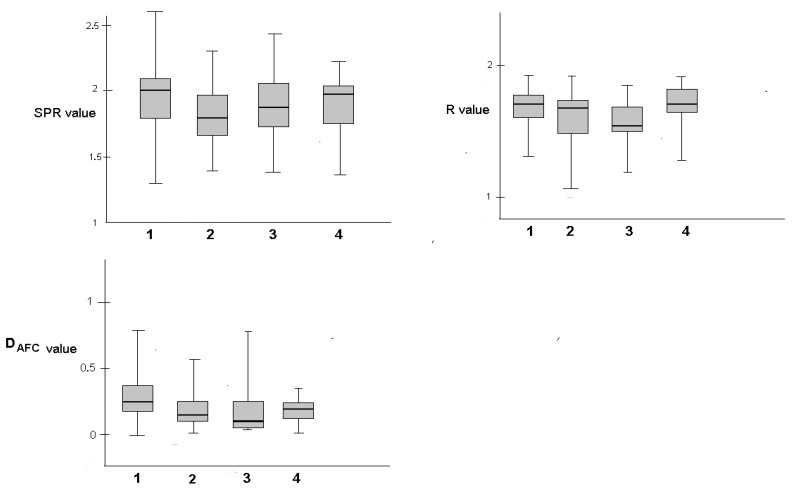
Graphical representation of values stratification for all introduced parameters and groups.

**Figure 5 diagnostics-14-01887-f005:**
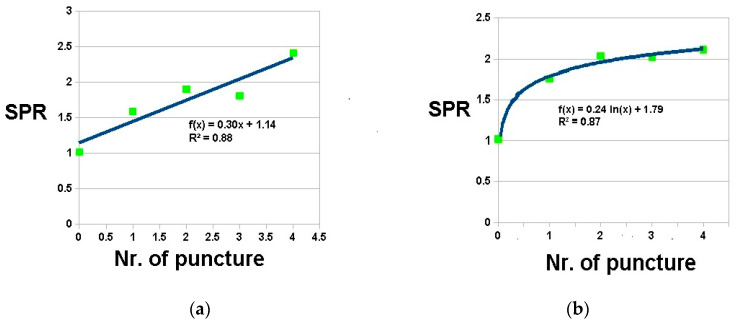
SPR as function of puncture number (**a**) linear trend; (**b**) logarithmic trend (R^2^-correlation coefficient of fit dependency).

**Table 1 diagnostics-14-01887-t001:** Summary of mean (+SD) values (values for last puncture are taken into account).

Group -Finding	1	2	3	4
R_m_ ^1^	1.71 (0.141 ^5^)	1.62 (0.26)	1.481 (0.184)	1.79 (0.211)
D_AFCm_ ^2^	0.29 (0.173)	0.167 (0.151)	0.158 (0.247)	0.18 (0.114)
SPR_m_ ^3^	1.99 (0.263)	1.787 (0.157)	1.64 (0.321)	1.97 (0.286)
N ^4^	24	19	10	7

^1^ mean value of R values for given group; ^2^ mean value of D_AFC_ values for given group; ^3^ mean value of SPR values for given group; ^4^ quantity of samples; ^5^ standard deviation of group values for given parameter.

**Table 2 diagnostics-14-01887-t002:** *t*-test for two independent mean between chosen groups.

SPR	1 > 3	1 > 2	2 > 3	4 > 2
t	3.195	2.74	−0.95	1.72
*p*, result	0.00154, signif. at *p* < 0.05	0.00446, signif. at *p* < 0.05	0.085, not. signif. at *p* < 0.05	0.045, signif. at *p* < 0.05
D_AFC_	1 >3	1 > 2	2 > 3	4 > 2
t	2.89	2.34	0.159	0.31
*p*, result	0.034, signif. at *p* < 0.05	0.0093, signif. at *p* < 0.05	0.325, not signif. at *p* < 0.05	0.378, not signif. at *p* < 0.05

## Data Availability

The data presented in this study are available on request from the corresponding author.

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
