# Peer review of "Near-Infrared On-Site Evaluation (NOSE) Examination of EBUS/EUSb Samples—A New Method for Sample Adequacy Evaluation"

_diagnostics, 2024, doi:10.3390/diagnostics14171887_

Round 1

Reviewer 1 Report

Comments and Suggestions for Authors

Dear All,

Thank you for allowing me to peer review this manuscript on a new potential tool for sample adequacy evaluation acquired during EBUS procedures. The authors have provided a detailed technical description of the Near Infrared Onsite Evaluation (NOSE) applied to the evaluation of biopsy samples obtained during EBUS, as an alternative to ROSE, which is often associated with costs and logistical limitations. The authors also validated this technique on a cohort of 60 patients. However, in my opinion, before this article can be considered for publication in Diagnostics, some major/minor revisions are necessary:

  • Page 1, Line 2: In the title, it would be better to write “Near Infrared Onsite Evaluation (NOSE)” rather than just NOSE.
  • Page 1, Line 11: The abstract primarily contains a technical description of NOSE and the conclusions, but it lacks the materials/methods and results sections.
  • Page 1, Line 13: Replace “ROSE” with “Rapid Onsite Evaluation (ROSE)”.
  • Page 2, Line 56: Replace “MOSE” and “ROSE” with “(MOSE)” and “(ROSE)”.
  • Page 3, Line 123: In the materials and methods section, information regarding the study design, inclusion/exclusion criteria, study outcomes with definitions, and references used are missing.
  • Page 6, Line 217: It would be useful to include a table with the baseline characteristics of the included patients.
  • Page 8, Line 250: Expand the discussion section to include a more in-depth comparison between NOSE and other existing techniques, highlighting the advantages and limitations of each, as well as potential clinical implications and future research directions.

Reviewer 2 Report

Comments and Suggestions for Authors

Thank you for inviting me to review the work "NOSE examination of EBUS/EUSb samples – a new method of sample adequacy evaluation". This is a very interesting work on the technique of near infrared onsite evaluation consisting of spectral measurement of near infrared radiation transmitted through evaluated material. The introduction is comprehensive and very well written, the methods and results are complete and described in an accessible way - the conclusions of the work are consistent with the results, and the discussion is conducted in a very interesting way. I have no comments on the above work and I congratulate the authors - the only thing that needs to be improved in the work are the issues of editing the text in some places - because there are a few spaces there or the text is simply not continuous because when applied to the template it can be seen that it sometimes falls apart - but the content is correct. The work is very good and these are small details during editing.

Round 2

Reviewer 1 Report

Comments and Suggestions for Authors

Dear All, 

the revised paper could be taken into consideration for a publication in diagnostics.